# Effect of Laser Peening on the Corrosion Properties of 304L Stainless Steel

**DOI:** 10.3390/ma16020804

**Published:** 2023-01-13

**Authors:** Young-Ran Yoo, Seung-Heon Choi, Young-Sik Kim

**Affiliations:** 1Materials Research Centre for Energy and Clean Technology, Andong National University, 1375 Gyeongdong-ro, Andong 36729, Republic of Korea; 2School of Materials Science and Engineering, Andong National University, 1375 Gyeongdong-ro, Andong 36729, Republic of Korea

**Keywords:** laser peening, stainless steel, passivation, intergranular corrosion, cross-section

## Abstract

Dry canisters used in nuclear power plants can be subject to localized corrosion, including stress corrosion cracking. External and residual tensile stress can facilitate the occurrence of stress corrosion cracking. Residual stress can arise from welding and plastic deformation. Mitigation methods of residual stress depend upon the energy used and include laser peening, ultrasonic peening, ultrasonic nanocrystal surface modification, shot peening, or water jet peening. Among these, laser peening technology irradiates a continuous laser beam on the surface of metals and alloys at short intervals to add compressive residual stress as a shock wave is caused. This research studied the effect of laser peening with/without a thin aluminum layer on the corrosion properties of welded 304L stainless steel. The intergranular corrosion rate of the laser-peened specimen was a little faster than the rate of the non-peened specimen. However, laser peening enhanced the polarization properties of the cross-section of 304L stainless steel, while the properties of the surface were reduced by laser peening. This behavior was discussed on the basis of the microstructure and residual stress.

## 1. Introduction

When metals and alloys are stressed beyond their elastic limits, they can be plastically deformed, and residual stress remains on metals and alloys. There are three reasons for this stress forming: When metals and alloys are cooled from a high temperature, there is often a thermal difference during cooling. The varying thermal difference develops non-uniform stress. In addition, phase transformation and mechanical processing can induce residual stress in metals and alloys. This residual stress facilitates the cracking and corrosion of metals and alloys. High residual stress can be generated near weldment due to the non-uniform thermal gradient formed from localized faster heating and cooling during the welding process [1,2].

Generally, mitigation methods for residual stress include post-welding heat treatment, low-temperature stress relief, mechanical stress relief, and surface modification methods. Of these methods, surface modification methods depend upon the energy used and include laser peening [3], ultrasonic peening [4], ultrasonic nanocrystal surface modification [5], shot peening [6], and water jet peening [7].

Laser peening is a technology that irradiates a continuous laser beam on the surface of metals and alloys at short intervals to add compressive residual stress as a shock wave is caused [8]. When laser peening is performed, plasma is concentrated on the surface using a transparent overlay (water [9] or glass [10] layer), and absorption layers (Al [11,12], vinyl tape [13,14]) are used to form effective compressive residual stress. When laser peening is performed by positioning an absorption layer that maintains the high quality of the surface and effectively maintains surface melting and energy conversion, compressive residual stress is formed [15] and grain refinement occurs [16], which leads to microstructural changes due to increasing dislocation density and strain [17]. This influences intergranular corrosion, corrosion fatigue, and SCC properties. We recently reported the effect of laser peening on the microstructure of 304L stainless steel: Laser peening resulted in the increased roughness of the surface, refinement of the grain size, deformation of the inside, increased dislocation density, and increased hardness of 304L stainless steel [18].

Corrosion properties by surface modification have recently been studied as follows: Shot peening on the corrosion of aluminum alloy [19,20] and the corrosion properties of stainless steel [21,22]; laser peening on the corrosion of Alloy 600 [23], the corrosion of 304 stainless steel [24], the corrosion of stainless steels [25,26,27,28,29], the corrosion of aluminum alloy [30], and the corrosion of magnesium alloy [31]; UNSM on the corrosion of nickel base alloys [32,33,34] and the corrosion of stainless steels [35,36,37,38]; and comparison between laser peening, shot peening, and ultrasonic peening [8]. Most recent research efforts on surface modification methods have shown beneficial effects on the corrosion of various alloys [19,20,21,22,23,24,25,26,27,28,29,30,31,32,33,34,35,36,37,38]. In some cases, detrimental effects of surface modification methods can be induced. For example, a high static load can degrade corrosion resistance [34]. According to a recent report by our group, ultrasonic nanocrystal surface modification increases corrosion resistance, but beyond the critical static load, corrosion resistance decreases as the repeated striking of the surface creates an overlapped wave which acts as the initiation site of corrosion [34]. As described above, surface modification methods including laser peening increase corrosion properties in some cases, but degrade resistance in other cases. That is, it is not clear how surface modification affects corrosion resistance.

In this study, we made laser-peened 304L stainless steel, which has a compressive residual stress of over 1 mm, evaluated intergranular corrosion properties and polarization behavior, and measured residual stress. Resistances were discussed and the effect of laser peening on corrosion properties was elucidated.

## 2. Experimental Methods

### 2.1. Specimen

The specimen used in this work was commercial 304L stainless steel; Table 1 shows the chemical composition of 304L stainless steel and its filler metal [18]. The thickness of the specimen was 25 mm. The specimen was welded using the gas tungsten arc welding (GTAW) method by the following conditions (see Table 2) [18]. Table 3 shows the designation of the specimen’s condition [18].

### 2.2. Laser Peening (LP)

The laser peening process employed in this study was well described elsewhere: Figure 1 shows the schematic diagram of laser peening and Table 4 summarizes the laser peening conditions [18]. The equipment for laser peening was self-manufactured using an Nd-YAG laser to protect the steel’s surface during the laser peening process and Al tape was used. The laser energy used was 4.4 J and a water layer of 1~2 mm was dynamically overlaid. The laser incident beam angle was 18°.

### 2.3. Qualitative Degree of Sensitization Measurement: ASTM A262 Pr. A

The specimens were cut to sizes of 15 × 15 × 10 mm. They were connected to an insulated electrical lead wire and mounted using an epoxy. Then, the cross-section of the specimen was prepared through grinding and polishing using #2000 SiC paper and diamond paste (3 μm), respectively. Specimens were insulated with a resin, except for an area of 1 cm^2^. The specimen was anodically dissolved for 90 s at 1 A/cm^2^ in 10% oxalic acid. After ultrasonic cleaning, the sample was observed using an optical microscope and the degree of sensitization was determined as shown in the standard [39].

### 2.4. Quantitative Degree of Sensitization Measurement: DL-EPR Test

The specimens were cut to sizes of 15 × 15 × 10 mm. They were connected to an insulated electrical lead wire and mounted using an epoxy. Then, the cross-section of the specimen was prepared through grinding and polishing using #2000 SiC paper and diamond paste (3 μm), respectively. The polished cross-section, except for an area of 0.09 cm^2^, was electrically insulated with a resin. According to ASTM G108, a double loop-electrochemical potentiokinetic reactivation (DL-EPR) test was performed [40]. The test solution was 0.5 M H_2_SO_4_ + 0.01 M KSCN at 30 °C and was deaerated at a rate of 200 mL N_2_/min for 30 min. A DL-EPR test was performed using a potentiostat (Interface 1000, Gamry Instruments, Warminster, PA, USA). Pt wire and a saturated calomel electrode were used as the counter and reference electrodes, respectively. Anodic scan to vertex potential (+400 mV(SCE)) and reactivation were swept, and the scan rate was at a rate of 1.677 mV/s. According to the standard [39], the ratio of Ir/Ia was determined, indicating the degree of sensitization (DOS). 

After the test, the specimen was taken out and ultrasonically cleaned, and the cross-section was observed using an optical microscope (AXIOTECH 100 HD, ZEISS, Oberkochen, Germany).

### 2.5. Intergranular Corrosion Rate Measurement: ASTM A262 Pr. C

The specimens were cut to sizes of 15 × 15 × 10 mm and aged for 1 h at 675 °C. Then, the specimen, except the peened surface, was ground using SiC paper from #120 to #2000. An immersion test in 65% HNO_3_ at boiling temperature was performed according to ASTM A262 Pr. C [39]; each test time took 3 h due to the possibility of severe corrosion on the peened surface. The tests were repeated five times. The intergranular corrosion rate was obtained from the weight loss and intergranular corrosion was confirmed by field emission-SEM (MIRA3 XMH, Tescan, Brno, Czech Republic) of the corroded cross-section.

### 2.6. Anodic Polarization Test

The specimens were cut to sizes of 15 × 15 × 10 mm. The specimens were connected to an insulated electrical lead wire. The non-peened specimen was mounted using an epoxy and ground using #2000 SiC paper, but the peened specimen was insulated with an epoxy resin. An area of 0.09 cm^2^ was exposed to a test solution. A potentiostat (Interface 1000, Gamry Instruments, Warminster, PA, USA) was used and a saturated calomel electrode for the reference electrode and Pt wire for the counter electrode were used. The test solution was 1% NaCl at 30 °C and deaerated at a rate of 200 mL N_2_/min for 30 min. After the specimen was immersed into Avesta cell, it was cathodically polarized at −200 mV from open circuit potential for 10 min to remove the surface oxide and then maintained at open circuit potential for 10 min. After the corrosion potential was measured, the polarization curve was obtained from −100 mV than the corrosion potential. The scan rate was 0.33 mV/s [41].

### 2.7. AC Impedance Test

The specimens were cut to sizes of 15 × 15 × 10 mm. The specimen and potentiostat used were the same as with the anodic polarization test. The test solution was 1% NaCl at 30 °C and deaerated at a rate of 200 mL N_2_/min for 30 min. The frequency range was 0.01~10,000 Hz, and an AC impedance test was performed at the corrosion potential. Polarization resistance was calculated using an equivalent circuit of Randles model [42].

### 2.8. Residual Stress Measurement

The residual stress was performed via a hole drilling method (RS-200 Assembly, VMM, USA). A strain gauge was attached and holes were fabricated by using a drilling device. The residual stresses released during the drilling were measured.

## 3. Results and Discussion

### 3.1. Effect of Laser Peening on the Intergranular Corrosion of 304L Stainless Steel

Figure 2 shows the effect of laser peening on intergranular corrosion by ASTM A262 Pr. A [39] for the outermost area of the cross-section of base metal, HAZ (heat affected zone), and weldment of 304L stainless steel. In the case of the base metal, the grain boundaries looked like the etched microstructure. In the case of the heat-affected zone, the grains were grown, and the grain boundaries also looked like the etched microstructure. However, in the case of weldment, a dendritic structure was observed, and the dendrite grew along the direction in which heat escapes. Compared to the sensitization of the base metal, the effect of laser peening on the sensitization of the cross-section of 304L stainless steel was slight, but the outermost area by laser peening showed a refined microstructure and more corrosion, regardless of the area. 

Figure 3 shows the effect of laser peening on the average grain size by ASTM E1382 [43]. Figure 3a shows the base metal, while Figure 3b shows the HAZ area. The average grain size was measured three times and the standard deviation was calculated. Regardless of the observed area, the laser peening treatment refined the grain size. These refinements were due to the outermost area of the laser peening. However, the average grain size of the HAZ area was greater than the size of the base metal, regardless of the peening condition. When the laser peening was forced to the surface, the grain size of the outermost area was refined, and this refinement yielded a partly smaller grain size for the observed area. Note that the refinement of the outermost area needed to be much more intensive than that of the observed area [18]. In addition, the average grain size of the HAZ area was greater than the size of the base metal due to the grain growth of the HAZ area by welding.

Figure 4 shows the effect of laser peening on the DL-EPR curve and DOS of the outermost area of the cross-section of 304L by a double loop-electrochemical potentiokinetic reactivation test. The DOS of base metal 304LB was 0.00003, but that of 304LB peened by laser without Al coating was 0.00641, while that of 304LB peened by laser with Al coating was 0.0002. That is, laser peening increased the degree of sensitization. The DOS of the HAZ area of 304LW was 0.00095, but that of the HAZ area of 304LW peened by laser without Al coating was 0.00735, while that of the HAZ area of 304LW peened by laser with Al coating was 0.00316. That is, laser peening also increased sensitization. The DOS of the weldment area of 304LW was 0.00104, but that of the weldment area of 304LW peened by laser without Al coating was 0.00614, while that of the weldment area of 304LW peened by laser with Al coating was 0.00285. That is, laser peening also increased the degree of sensitization. This behavior was due to the increased grain boundary area by the laser peening and showed a similar trend to those of the qualitative degree of sensitization, as shown in Figure 2.

Figure 5 shows the optical micrographs of the cross-section of 304L after the DL-EPR test. Compared to the photos of the base metal, the DL-EPR test corroded the outmost area of the laser-peened cross-section relatively more. However, the effects of Al-coating on the laser peening process were difficult to differentiate from each other.

Figure 6 shows the effect of laser peening on the intergranular corrosion rate of 304L obtained from the immersion test in boiling 65% HNO_3_ [39]. The maximum allowable corrosion rate was referred from ISO 12732 [44]. Figure 6a shows that laser peening treatment of the base metal increased the intergranular corrosion rate, regardless of Al coating. These increased results were similar to those of the qualitative intergranular corrosion test in Figure 2 and the quantitative intergranular test in Figure 4. Figure 6b shows that laser peening treatment of the welded specimen also increased the intergranular corrosion rate, regardless of Al coating. These results were similar to those of the qualitative intergranular corrosion test in Figure 2 and the quantitative intergranular test in Figure 4. Comparing the results of the base metal and welded specimen, the welding process slightly increased the intergranular corrosion rate, but the effect of laser peening of the welded specimen was a little stronger than that of the base metal. Table 5 shows the relationship between the degree of sensitization and intergranular corrosion (IGC) rate of 304L stainless steel by laser peening. If the relationships between them, except that of 304LB-L-WC, are plotted, the trend equation is obtained as ‘IGC rate, mm/y = 52.4 × (DOS) + 0.1’; its determination coefficient was calculated as 0.9919.

Figure 7 shows the corroded grain boundaries in the cross-section of non-peened and peened 304L stainless steel after the immersion test for 15 h in boiling 65% HNO_3_. In the case of 304L base metal, the grain boundaries were selectively corroded, regardless of the area. However, in the case of laser-peened specimens, the grain boundaries of the inside coarse grains were selectively corroded, although, the grain boundaries refined by laser peening were selectively corroded. Therefore, the intergranular corrosion rate of the laser-peened specimen was increased over that of the non-peened specimen. We recently reported the detrimental and beneficial effects of surface modification technology; in the case of non-sensitized 316L stainless steel, the effect of ultrasonic nano-crystal surface modification treatment on intergranular corrosion was detrimental, but this treatment on slightly sensitized 316L stainless steel showed a beneficial effect on intergranular corrosion. This behavior was due to reduced carbon segregation, grain refinement, and compressive residual stress [36].

### 3.2. Effect of Laser Peening on the Polarization Behavior of 304L Stainless Steel

Figure 8 shows the effect of laser peening on the polarization curves on the surface area of 304L stainless steel in deaerated 1% NaCl at 30 °C. Figure 8a shows that the transpassive potential of the 304L base metal was 0.985 V (SCE) and the stable passive current density was obtained, but the pitting potentials of the laser-peened 304LB-L-NC and 304LB-L-WC were 0.25 and 0.107 V (SCE), respectively. In the case of the HAZ area, the pitting potentials of the non-peened, peened without coating, and peened specimens with coating were 0.646, 0.056, and 0.402 V (SCE), respectively (Figure 8b). Moreover, in the case of weldment, the pitting potentials of the non-peened, peened without coating, and peened specimens with coating were 0.795, 0.086, and 0.268 V (SCE), respectively, as shown in Figure 8c. In summary, in the case of the polarization behavior of the surface, it is well known that the welding process reduces resistance due to sensitization and microstructural change [18]; laser peening decreased the pitting resistance, which was related to the surface roughness, as described elsewhere [18]. Irrespective of Al coating, the laser peening roughened the surface [18], and a high static load in UNSM could degrade the corrosion resistance as a high static load and repeated process can create an overlapped wave [33].

Figure 9 shows the effect of laser peening on the polarization curves in the cross-sectional area of 304L in deaerated 1% NaCl at 30 °C. Figure 9a shows that the pitting potential of the 304L base metal was 0.212 V (SCE), but the pitting potentials of the laser-peened 304LB-L-NC and 304LB-L-WC were 0.23 and 0.428 V (SCE), respectively. Laser peening also reduced the passive current densities. In the case of the HAZ area, the pitting potentials of the non-peened, peened without coating, and peened specimens with coating were 0.222, 0.235, and 0.250 V (SCE), respectively, as in Figure 9b. Moreover, in the case of weldment, the pitting potentials of the non-peened, peened without coating, and peened specimens with coating were 0.065, 0.326, and 0.375 V (SCE), respectively, as shown in Figure 9c. In addition, laser peening decreased the passive current densities. In summary, laser peening on the outermost area of the cross-section of 304L stainless steel enhanced the polarization properties, irrespective of Al coating, which is related to the microstructural change by laser peening. As recently reported, laser peening refined the grain size, deformed the inside, and increased the dislocation density [18]. Similar effects in UNSM-treated Alloy 600 were discussed [33].

The corrosion current density (i_R_), corrosion potential (E_R_), and pitting potential (E_P_) of the surface and cross-section of 304L stainless steel are summarized in Table 6 and Table 7, respectively. In the case of the surface, laser peening increased the corrosion current density and reduced the pitting potential, while laser peening decreased the corrosion current density and increased the pitting potential of the cross-section (corrosion current density was obtained through Tafel extrapolation analysis).

Table 8 summarizes the residual stress measured on the surface and at 1 mm depth of 304L stainless steel via a hole drilling method: The surface and 1 mm depth of non-peened 304L stainless steel showed tensile residual stress, but compressive residual stress of the peened specimen was formed, regardless of measuring areas.

Therefore, it can be elucidated that laser peening treatment can refine grain size [18] and induce compressive residual stress, thereby improving the passivation properties of 304L stainless steel. However, it should be noted that the pitting resistance of the surface by laser peening was reduced due to the increased irregularity including an overlapped surface and increased corrosion initiate sites by laser peening [34].

A comparison of the polarization curves of 304L stainless steel in Figure 8 and Figure 9 shows that the pitting potentials of the cross-section were much lower than those of the surface. As the surface was rougher than the cross-section, this was an interesting result. Figure 10 shows the corrosion morphology of the cross-section after an anodic polarization test of 304LB. It was electrochemically etched using 10% oxalic acid solution. The optical micrograph (Figure 10a) shows the mechanical flow lines by the rolling process. An SEM micrograph (Figure 10b) shows the pits in the mechanical flow line area, including grain boundaries. Thus, the mechanical flow line formed by the rolling process seemed to act as the pitting initiation site and thus this flow reduces the pitting corrosion resistance of 304L stainless steel, as shown in Figure 10c.

Figure 11 shows the effect of laser peening on the AC Impedance behavior of the cross-section of 304L stainless steel in deaerated 1% NaCl at 30 °C. The polarization resistance of the 304L base metal was 1230 kΩ, but the resistances of the laser-peened 304LB-L-NC and 304LB-L-WC were 2250 and 1800 kΩ, respectively. In the case of the HAZ area, the resistance of the non-peened specimen was 341 kΩ, but the resistances of the laser-peened 304LW-H-L-NC and 304LW-H-L-WC were 1070 and 1730 kΩ, respectively. In the case of the weldment area, the resistance of the non-peened specimen was 131 kΩ, but the resistances of the laser-peened 304LW-W-L-NC and 304LW-W-L-WC were 1900 and 1510 kΩ, respectively. In summary, regardless of base metal and welded areas, the laser peening enhanced the passive properties, and this was consistent with the above polarization test.

## 4. Conclusions

In this study, the effect of laser peening on the intergranular corrosion and polarization behavior of 304L stainless steel was evaluated. The specimens with a compressive residual stress of over 1 mm were made by laser peening treatment. The following conclusions are drawn:(1)The intergranular corrosion rate of the laser-peened 304L stainless steel was a little faster than the rate of the non-peened specimen. The increased area of grain boundaries by laser peening reduced the intergranular corrosion resistance, while an Al coating layer did not influence resistance.(2)Laser peening of 304L stainless steel enhanced the polarization properties of the cross-section. This behavior was related to the grain refinement and compressive residual stress induced by laser peening treatment, irrespective of Al coating.

## Figures and Tables

**Figure 1 materials-16-00804-f001:**
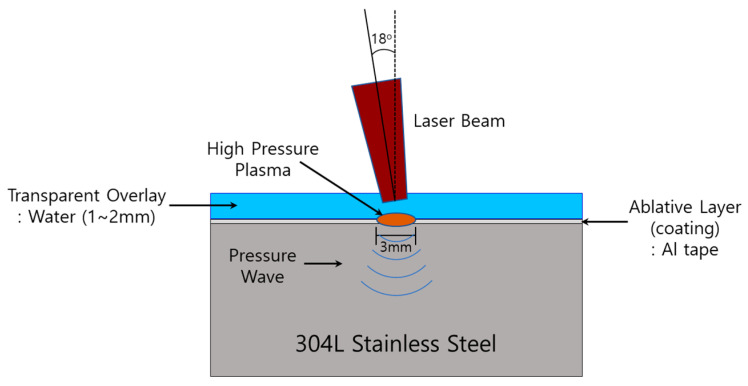
Schematic of laser peening process [18].

**Figure 2 materials-16-00804-f002:**
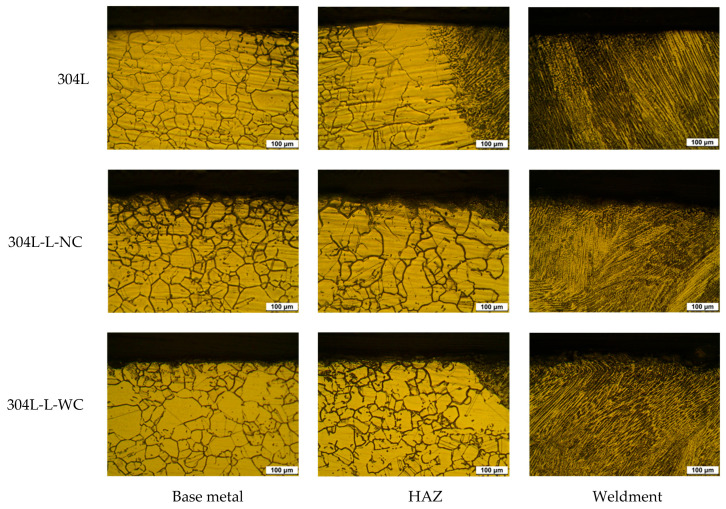
Effect of laser peening on the degree of sensitization obtained by ASTM A262 Pr. A for the outermost area of the cross-section of 304L stainless steel (OM, ×200).

**Figure 3 materials-16-00804-f003:**
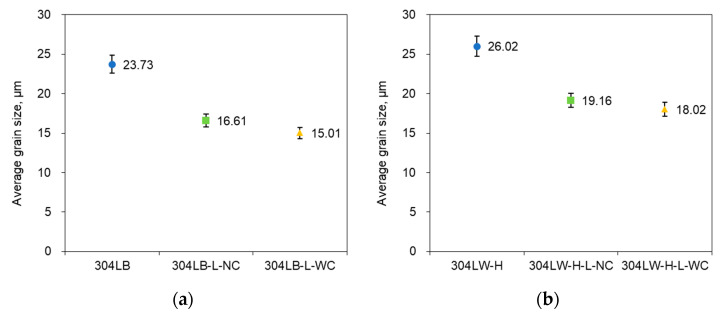
Effect of laser peening on grain size by ASTM E1382; (**a**) Base metal, (**b**) HAZ area.

**Figure 4 materials-16-00804-f004:**
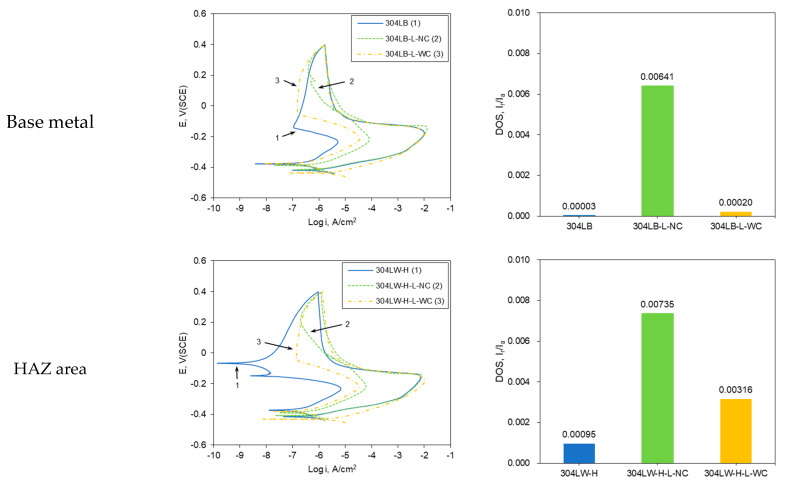
Effect of laser peening on the degree of sensitization on the outermost area of the cross-section of 304L by DL-EPR test.

**Figure 5 materials-16-00804-f005:**
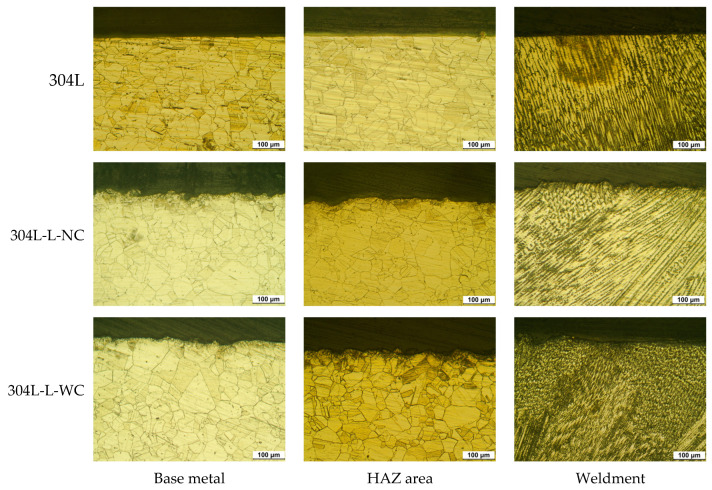
Cross-sectional appearance of 304L after DL-EPR test (OM, ×200).

**Figure 6 materials-16-00804-f006:**
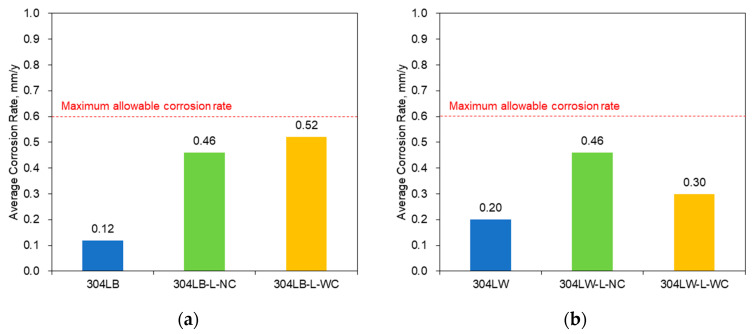
Effect of laser peening on the intergranular corrosion rate of 304L obtained from the immersion test in boiling 65% HNO_3_; (**a**) Base metal, (**b**) Welded specimen.

**Figure 7 materials-16-00804-f007:**
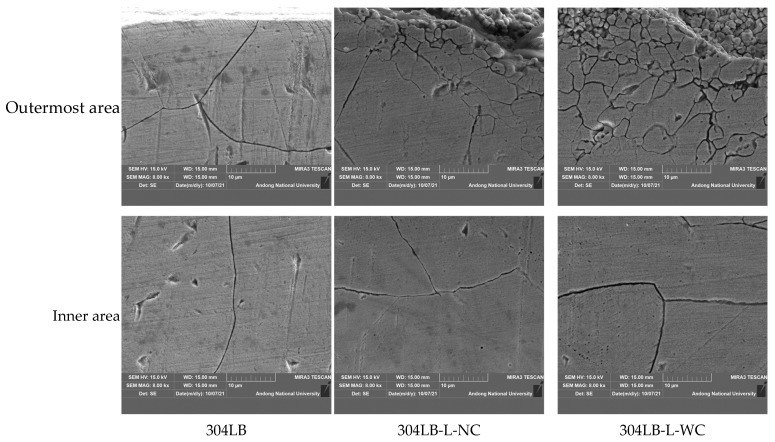
Corroded grain boundaries (SEM, ×8000) in the cross-section of 304L stainless steel after the immersion test for 15 h in a boiling 65% HNO_3._

**Figure 8 materials-16-00804-f008:**
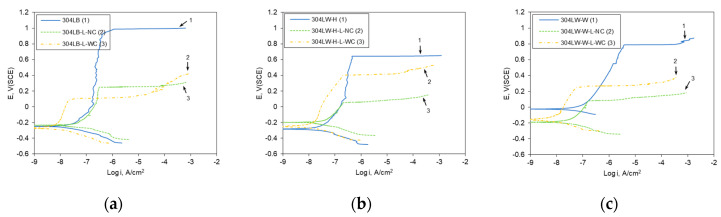
Effect of laser peening on the polarization curves of 304L-surface area (30 °C, deaerated 1% NaCl); (**a**) Base metal, (**b**) HAZ area, (**c**) Weldment.

**Figure 9 materials-16-00804-f009:**
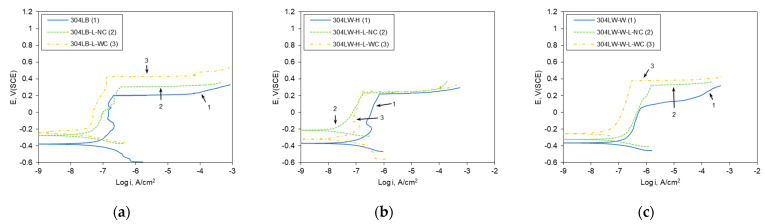
Effect of laser peening on the polarization curves of 304L-cross-section area (30 °C, deaerated 1% NaCl); (**a**) Base metal, (**b**) HAZ area, (**c**) Weldment.

**Figure 10 materials-16-00804-f010:**
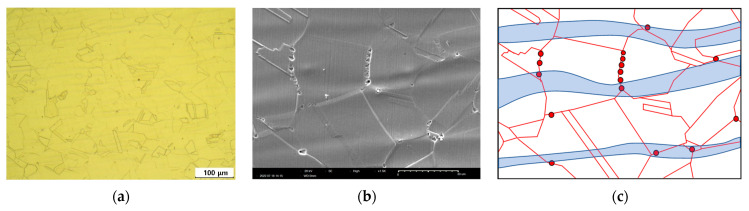
Corrosion morphologies of cross-section after anodic polarization test of 304LB (10% Oxalic acid, (**a**) OM (×100), (**b**) SEM (×1500)), and (**c**) schematic diagram (blue band is mechanical flow and red points are pits).

**Figure 11 materials-16-00804-f011:**
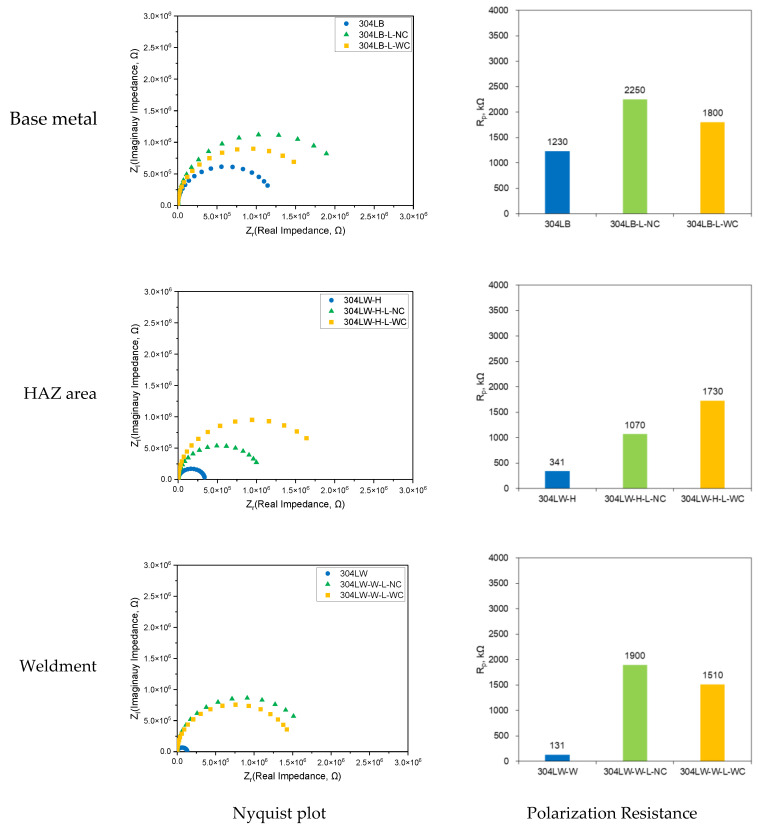
Effect of laser peening on the AC Impedance of 304L-cross-section area (30 °C, deaerated 1% NaCl).

**Table 1 materials-16-00804-t001:** Chemical composition of 304L stainless steel and filler metal (wt %) [18].

	C	Cr	Ni	Mn	Si	Cu	Mo	Co	P	N	S	Cb + Ta	Fe
304L	0.02	18.6	9.6	1.65	0.47	-	-	0.03	0.022	0.07	0.03	-	Bal.
ER308L	0.015	19.81	9.84	1.691	0.351	0.115	0.046	0.030	0.024	0.041	0.03	0.008	Bal.

**Table 2 materials-16-00804-t002:** Welding conditions of the experimental specimen [18].

Welding Process	Current (A)	Voltage (V)	Speed (cm/min)	Shield Gas (%)	Groove Angle (°)	Welding Electrode
GTAW	245~250	14~15	9~10	Ar. 99.9	15	ER308L (Dia. 0.9 mm wire)

**Table 3 materials-16-00804-t003:** Designation of the experimental specimen [18].

Alloy		Non-Peened	Laser Peening	
			Non-Coated	With Coating
304L	Base metal	304LB	304LB-L-NC	304LB-L-WC
	HAZ area	304LW-H	304LW-H-L-NC	304LW-H-L-WC
	Weldment	304LW-W	304LW-W-L-NC	304LW-W-L-WC

**Table 4 materials-16-00804-t004:** Conditions of laser peening treatment [18].

Laser Type	Laser Energy (J)	Laser Spot Diameter (mm)	Laser Overlay (%)	Transparent Overlay	Laser Incident Beam Angle (°)	Coating
Nd-YAG (1064 nm, IR)	4.4	3	50	Water (1~2 mm)	18	Al tape

**Table 5 materials-16-00804-t005:** The relationship between the DOS and IGC rate of 304L stainless steel by laser peening.

	304LB	304LB-L-NC	304LB-L-WC	304LW-W	304LW-W-L-NC	304LW-W-L-WC
DOS, I_r_/I_a_	0.00003	0.00641	0.0002	0.00104	0.00614	0.00396
IGC rate, mm/y	0.12	0.46	0.52	0.2	0.46	0.3

**Table 6 materials-16-00804-t006:** The corrosion factors of the surface of 304L stainless steel obtained from Tafel extrapolation.

Surface Area	Non-Peened 304L			304L-L-NC			304L-L-WC		
	i_R_, nA/cm^2^	E_R_, V(SCE)	E_P_, V(SCE)	i_R_, nA/cm^2^	E_R_, V(SCE)	E_P_, V(SCE)	i_R_, nA/cm^2^	E_R_, V(SCE)	E_P_, V(SCE)
Base metal	12.28	−0.248	0.985	26.48	−0.232	0.25	4.03	−0.266	0.107
HAZ	11.91	−0.283	0.646	23.06	−0.197	0.056	5.83	−0.247	0.402
Weldment	24.34	−0.020	0.795	23.99	−0.188	0.086	3.29	−0.155	0.268

**Table 7 materials-16-00804-t007:** The corrosion factors of the cross-section of 304L stainless steel obtained from Tafel extrapolation.

Cross-Section Area	Non-Peened 304L			304L-L-NC			304L-L-WC		
	i_R_, nA/cm^2^	E_R_, V(SCE)	E_P_, V(SCE)	i_R_, nA/cm^2^	E_R_, V(SCE)	E_P_, V(SCE)	i_R_, nA/cm^2^	E_R_, V(SCE)	E_P_, V(SCE)
Base metal	42.97	−0.378	0.212	17.94	−0.275	0.23	7.89	−0.245	0.428
HAZ	58.00	−0.367	0.222	10.33	−0.213	0.235	24.73	−0.317	0.250
Weldment	46.31	−0.364	0.065	44.82	−0.325	0.326	16.48	−0.252	0.375

**Table 8 materials-16-00804-t008:** Residual stress on the surface and at 1 mm depth of 304L stainless steel.

Residual Stress, MPa	Non-Peened 304L		304L-L-NC		304L-L-WC	
	Surface	1 mm-Depth	Surface	1 mm-Depth	Surface	1 mm-Depth
Base metal	92	14	−786	−128	−794	−131
HAZ	335	41	−497	−55	−565	−56
Weldment	11	119	−532	−118	−629	−229

## Data Availability

Not applicable.

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
