# Peer review of "Effect of Laser Peening on the Corrosion Properties of 304L Stainless Steel"

_materials, 2023, doi:10.3390/ma16020804_

Round 1

Reviewer 1 Report

In this paper the authors studied the effect of laser peening (an impartment of beneficial residual stresses) on the corrosion properties of gas tungsten arc welded stainless steel. Both the uncoated and the Al-coated specimens were characterized. The article is a follow-up of a previously published work (https://doi.org/10.3390/ma15113947, reference [39]). The current paper is well-researched and contains a lot of experimental corrosion resistance results. Different techniques have been combined to yield a detailed characterization (double loop EPR, boiling HNO3 test, anodic polarization, EIS). The characterization is as complete as possible. The paper is definitely worthy of publication in Materials subject to revision.

1.It would help to provide more details on the Al coating. How were the steels coated?

2.The abstract is very general and copies the introduction. A concrete specification of the materials studied and the results obtained is required. You should specify the corrosion rates measured, and list the most important conclusions. In the abstract, be as specific as possible.

3.The authors refer to their previous paper (ref. [39]) for laser peening details. Nevertheless, the experimental details should also be provided here as it is a standalone paper.

4.Fig. 3 shows the average grain size number of the welded and laser-peened specimen. A standard deviation should be provided in the figure. Furthermore, it is recommended to calculate the physical grain size itself (in micrometers) from the data or provide an estimate. It may not be directly obvious to every reader that a smaller grain size number means a larger grain size.

5.Arrows should be included in Fig. 4 to indicate the direction of polarization. Furthermore, colors should be used consistently throughout the manuscript. If green is used for the uncoated laser-peened specimen it should be green in both left and right figures. The same holds for coated laser-peened specimen (yellow). Otherwise, the readers might get confused. Correct also the color coding of Figures 8 and 9.

6.The polarization curves (Figs. 8 and 9) should be analyzed by Tafel extrapolation. Corrosion currents and corrosion potentials should be obtained, tabulated, and compared. The pitting potential should also be included in the same table for the sake of completeness.

7.Have you tried to measure an open circuit potential before the anodic polarization? If so, what were the results?

Author Response

We attached the file.

Reviewer 2 Report

Comments (Manuscript ID: materials-2056251):

In this paper, the intergranular corrosion properties and polarization behavior of laser peened 304L stainless steel were discussed. The content in this study is detailed and logical. Nevertheless the manuscript can be further improved, and there are some questions and suggestions for the authors.

Major revision:

1. All histograms in this paper have no error bars. If there was any error in this experiment, it needs to be marked on the charts.

2. The names of the sample in Figure 8 & Figure 9 are inconsistent with the description in part 3.2, please confirm carefully.

3. In line 281, page 10, the author claims “…therefore improved the passivation properties of 304L stainless steel”. However, according to the polarization curves in Figure 8, the laser peening is not conducive to its corrosion resistance, please explain it further.

4. In paragraph 2, page 10, the presentation is very confusing. If the author wanted to demonstrate the difference in corrosion degree of surface and cross section caused by different roughness, a comparison of pitting morphologies under all conditions should be provided, rather than just a morphology of cross section of 304LB.

5. Please recheck the expression of Figure 11 in page 11, there have some mistakes about the name of samples, such as “but the resistances of the laser-peened 304LW-H-L-NC and 304LW-H-L-NC were (1,070 and 1,730) kΩ…”, and “but 308 the resistances of the laser-peened 304LW-W-L-NC and 304LW-W-L-NC were…”.

6. The front part of abstract is redundant, please further modified and refined it.

7. The introduction should clarify the importance and necessity of this research object, so it is suggested to supplemented the relevant contents.

Author Response

We attached the file.

Reviewer 3 Report

1. Experimental Methods: The experimental methods are relatively detailed, and the standard number used in the experiments is also given. How to get the parameters related to the LSP experiment? The photos of the experimental samples are not given.

2. Effect of Laser Peening on the Intergranular Corrosion of 304L Stainless Steel. For the purpose of discussing intergranular corrosion, it is necessary to give a specific discussion on the relationship between sensitization degree and intergranular corrosion.

3. In Fig. 3, is the average grain size of HAZ area before and after LSP larger than that of base metal or smaller?

4. The weld corrosion after LSP treatment is intensified, which requires more sufficient arguments。

5. It is desirable to mark the flow line and the corresponding starting pitting position in Figure 10 with colored lines.

Author Response

We attached the file.

Round 2

Reviewer 1 Report

Authors answered most of my comments. The paper can be accepted for publication.

Reviewer 3 Report

The authors have addressed my concerns. Now I would like to suggest it to be published.